# Dynamical Synergies of Multidigit Hand Prehension

**DOI:** 10.3390/s22114177

**Published:** 2022-05-31

**Authors:** Dingyi Pei, Parthan Olikkal, Tülay Adali, Ramana Vinjamuri

**Affiliations:** Department of Computer Science and Electrical Engineering, University of Maryland Baltimore County, Baltimore, MD 21250, USA; dpei1@umbc.edu (D.P.); polikka1@umbc.edu (P.O.); adali@umbc.edu (T.A.)

**Keywords:** dynamical synergies, kinematic synergies, principal component analysis, singular value decomposition, movement primitives, hand prehension

## Abstract

Hand prehension requires highly coordinated control of contact forces. The high-dimensional sensorimotor system of the human hand operates at ease, but poses several challenges when replicated in artificial hands. This paper investigates how the dynamical synergies, coordinated spatiotemporal patterns of contact forces, contribute to the hand grasp, and whether they could potentially capture the force primitives in a low-dimensional space. Ten right-handed subjects were recruited to grasp and hold mass-varied objects. The contact forces during this multidigit prehension were recorded using an instrumented grip glove. The dynamical synergies were derived using principal component analysis (PCA). The contact force patterns during the grasps were reconstructed using the first few synergies. The significance of the dynamical synergies, the influence of load forces and task configurations on the synergies were explained. This study also discussed the contribution of biomechanical constraints on the first few synergies and the current challenges and possible applications of the dynamical synergies in the design and control of exoskeletons. The integration of the dynamical synergies into exoskeletons will be realized in the near future.

## 1. Introduction

The human hand is a dexterous and sophisticated sensorimotor system, capable of performing complex motor functions. Currently, a large population of individuals is suffering from the loss of hand mobility, including amputations, stroke, and spinal cord injury. Loss of dexterity can significantly affect the level of autonomy and the capability of individuals to perform their activities of daily living (ADLs), and they need the compensation of assistive devices, prosthetics and exoskeletons. Current solutions can substitute the appearance and function of the limb and accomplish motor control by providing basic functions. However, two of the major limitations still hamper the completion and dexterity of the normal hand motor capabilities. One is the high-dimensional control and another is the sensory feedback [1,2].

It has been hypothesized that the central nervous system (CNS) is able to control the complex movements of the human hand by controlling synergies instead of controlling the individual joints or individual degrees of freedom (DoFs), thus reducing the computational burden [3]. Mathematically, by using linear and nonlinear dimensionality reduction methods and matrix factorization methods, the synergies were derived from different measurements of hand movements, such as kinematic synergies from joint angular velocities [4,5] postural synergies from hand postures [6,7], and dynamical synergies derived from the contact forces during the grip tasks [8,9]. Principal Components Analysis (PCA) has been successfully applied in the extraction of synergies by us and others [4,5,6,7,8,10], and evidence shows that a much lower-dimensional subspace of the hand DoFs space can efficiently characterize most of the motions. In the design and implementation of robotics, prosthetics and exoskeletons [7,11,12,13,14], synergies introduce an alternative control mechanism to simplify the complexity of control and improve efficiency and functionality.

Research is currently underway to investigate how to effectively reduce the redundancy to deliver sensory information back to the user for bidirectional (feedforward and feedback) prosthesis control of grasping and manipulation using synergies. Researchers have made early attempts to mimic human feedforward and feedback control mechanisms by developing mechanotactile interfaces that estimate the interaction forces in contact with the grasped object and vibrotactile feedback interface that estimates surface irregularities and avoids slippage [15]. It was observed that the dimensionality reduction achieved by synergies is not just limited to motor control, but also can be used to reduce a large number of sensory inputs to a small set of manageable and controllable representations that can directly benefit modeling the sensory feedback in robotics and prosthetics [16].

During a grasp, the hand generates the manipulative contact forces based on the task environment, the task constraints and the task requirements while maintaining a stable grasp [11,17,18]. The optimal contact forces must be not too small to cause slippage or not too large to damage either the hand or the object under grasp. Furthermore, how the contact force collected from a large group of tactile sensors is mapped, distributed and correlated seems quite important to help understand and estimate the performance of a prehension task [19,20]. Since such a grasp involves a larger number of DoFs that need to be controlled simultaneously, several studies have hypothesized and shown that this could also happen in a low-dimensional space [8,21,22], similar to kinematic synergies. Thus, this study explores whether the dynamical synergies can characterize coordinated contact force patterns and provide this low-dimensional space. Since the interaction of the motor and sensory function are important to help restore the hand function, it needs to be demonstrated that the dynamical synergies can provide generalized force strategies. 

In this paper, we focused primarily on how the low-dimensional representations of contact forces in the human hand—dynamical synergies i.e., the synergies derived from the contact forces in different object grasping tasks contribute to the coordination of contact forces from multiple hand areas, and how these patterns of coordination vary across the different weights of the objects, replicating our tasks in the activities of daily living.

## 2. Materials and Methods

### 2.1. Experimental Protocol

A total of ten right-handed, healthy subjects (four male and six female) were included in this experiment under the approved IRB protocol at the Stevens Institute of Technology. They were asked to sit in front of a table to perform four object grasping tasks—ball, door handle, bottle cap and water bottle corresponding to four typical hand grasp types from ADLs–whole hand grasp, hook grasp, precision grasp and cylindrical grasp, respectively. To investigate the effect of the weight of the object on the grip force, these objects—ball, door handle, bottle cap and water bottle—were prepared with four different weights (170, 320, 470 and 620 g). The surface of the objects was wrapped by the same kind of material to remove the bias induced by friction forces.

During each trial of the experiment, the subjects first placed their hands around the object as if they were grasping it without contact. The minimal forces at this point that could be due to the device noise were used as a baseline. After two seconds, the subjects were cued by an auditory beep to grasp, lift and steadily hold the object for four seconds. After hearing the stop cue, the subjects put the object back to the original place and withdrew their hands back to the initial position. The objects were grasped in random order, and 20 repetitions were conducted for each weight. The contact force during the whole period was recorded by a GripGlove (Tekscan, Boston, MA, USA), an instrumented glove embedded with sensors to measure contact forces.

### 2.2. Preprocessing

Data was recorded by Research Foot software (Tekscan, Boston, MA, USA) from GripGlove, as shown in Figure 1A. The force sensors were divided into 12 areas including upper palm (UP), lower palm (LP), five distal phalanges (T1, In1, M1, R1 and P1 represent thumb to pinky respectively), thumb proximal phalanx (T2) and four middle-proximal phalanges (In2, M2, R2 and P2 represent index to pinky respectively, as shown in Figure 1B). Two types of data were saved from the steady-hold period in this study. One was the average force within each of the above 12 areas (Figure 1C), and the other was a spatial matrix of raw forces of the hand, which was selected from the fourth second, represented as a pixelated spatial force map (Figure 1D).

### 2.3. Derivation of Dynamical Synergies

The contact force dynamics considered in this study were represented by the averaged force from the steady-hold period and the spatial force maps. The effects of the weight of the object were analyzed individually to investigate how it influences the movement dynamics and reconstruction of the force patterns.

We hypothesized that the force patterns can be modeled as a weighted linear combination [4] of a few dynamical synergies, and the first few synergies represent the most variance among diverse grasp force patterns. Here, principal component analysis (PCA) was performed using singular value decomposition (SVD) to extract the dynamical synergies as shown below:(1)V=U∑S
where V is the force matrix with dimensions m×n, where m is the number of hand grasps and n is the number of forces recorded from the hand areas. For the averaged force data, the force matrix contains 12 forces, calculated from 12 hand areas; for the spatial force maps, the forces are represented as concatenated pixels, where a total of 361 pixels were included. S contains the principal components (PCs), which are considered as the dynamical synergies. ∑ is a diagonal matrix (with singular values of $\lambda_{1},\;\lambda_{2},\;\lambda_{3},\;\ldots ,\;\lambda_{n}$) and the magnitude of the PCs were determined as W=U∑. 

Three-fourths of the grasping tasks were used to extract the dynamical synergies and the remaining one-fourth were used for testing the synergies in the reconstruction of force patterns, and it was evaluated with four-fold cross-validation. After the dynamical synergies were derived, the magnitude of the dynamical synergies for the testing data were calculated by least-squares approximation. The force patterns were reconstructed by recruiting a few top-order dynamical synergies. The reconstruction error between the recorded force pattern (F) and the reconstructed force pattern ($F^{\prime }$) was determined as follows:(2)$$err=\frac{\sum_{i}{\left(F_{i}-F_{i}^{\prime }\right)}^{2}}{\sum_{i}F_{i}}\;$$

## 3. Results

### 3.1. Reconstruction Error Achieved Using the Dynamical Synergies

The dynamical synergies were extracted from two types of force patterns—the averaged force from the steady-hold period and spatial force maps. Since the top-ranked PCs or the top-order dynamical synergies represent the most significant variance directions among all the forces involved in the hand grasps, the fraction of variance would help to determine the number of dynamical synergies that could be used for optimal force pattern reconstruction. The reconstruction error of testing data across ten subjects and the variance accounted are illustrated in Figure 2.

For the averaged force (Figure 2A), 12 PCs were extracted. The first two synergies accounted for over 90% of the variance, and the average reconstruction error reduced to 0.2. With the first four synergies the error further reduced below 0.1. The mean reconstruction errors for ten subjects for four weights are listed in Table 1 with standard deviation. For the spatial force maps, 361 synergies were extracted, and only the first 50 synergies are plotted in Figure 2B. The first synergy only accounted for 50% of the total variance and the first five synergies accounted for about 80% of the variance. The differences between the synergies obtained from averaged forces and spatial force maps are intuitive due to the larger dimensional space of the data in spatial force maps.

### 3.2. Reconstruction of Spatial Force Maps

The spatial force maps for four different grasps are shown in Figure 3. The force distribution varies for different grasp types. The whole hand grasp (ball) and cylindrical grasp (water bottle) are characterized by larger contact areas, which are mostly concentrated at the finger whereas the hook grasp (door handle) requires palm, thumb and fingers. For precision grasp (bottle cap), the dominant fingers are fewer, and the force is exerted and concentrated at the fingertips of thumb, index and middle fingers.

Each spatial force map consists of 361 pixels; thus, a total of 361 synergies was calculated by SVD. Figure 4 shows a comparison between the recorded and the reconstructed spatial force maps using different numbers of synergies. In this example, the spatial force map of a whole hand grasp from Subject 6 was reconstructed. The whole hand grasp was selected as an illustrative test task because most of the finger digits are recruited to perform this type of grasp. Especially the thumb, middle and ring fingers accounted for the highest force intensity. Although the spatial force maps could not be thoroughly restored by using only a few dynamical synergies, these synergies are able to capture the most dominant characteristics of prehension in the human hand. Additionally, as was shown in Figure 2B, the first few synergies accounted for almost 90% of the variance but the reconstruction errors were higher in the range of 0.31 ± 0.06. Using the first 50 synergies, the reconstruction error further reduced to 0.12 ± 0.03. Thus, it can be noted that reconstruction accuracy increases by recruiting more synergies. However, with the use of fewer synergies, it was possible to summarize the dominant characteristics of the force patterns (Figure 4), such as the thumb, index and middle fingertips, which are the dominant force zones in our ADLs.

Similar results were observed from the reconstruction patterns from varied weights, that only a few synergies could successfully recognize the most dominant areas, as illustrated in Figure 5. Within the same type of grasp, incrementing the weight of the object contributed not only to the increased contact area and more fingers recruited in grasping but also an increase in the force intensities. The grip zones enlarged from fingertips to whole fingers even to some parts of the palm with the addition of weights. For the whole hand grasp (Figure 5), by recruiting only two dynamical synergies, the reconstructed patterns could successfully capture the dominant grip areas.

## 4. Discussion

### 4.1. Significance of Dynamical Synergies

Previous studies by us and others have shown the anatomical correlation and significance of synergies in muscles, kinematics and postures [4,6,23,24]. It is hypothesized that the central nervous system (CNS) may generate and manipulate high-dimensional movements by recruiting synergies or movement primitives in a low-dimensional space [24,25,26]. Only a few studies reported on how the dynamical synergies contribute to force control during grasping in our activities of daily living. As far as we know, none of the studies have looked at spatial force maps. 

Different individuals grasp differently due to nature and nurture. Some finger digits adapt for a variety of grasps and can withstand higher prehension forces, such as thumb, index and middle finger, as observed in this study. Latash et al. [27,28] have shown that finger digits act synergistically in the force production tasks. The results in this paper also suggest that the prehension forces could be characterized by coordination patterns (addressed as dynamical synergies in this paper) and thus reduce the DoFs involved in cortical control to achieve the grasp forces. We hypothesize that the dynamical synergies could represent the primitives of hand prehension dynamics. In other words, the distribution of prehension forces can be considered as a linear superposition of synchronized dynamical synergies. Furthermore, by increasing the resolution of the pixels in the spatial force maps we can understand precise contact areas and force points that can be of significant benefit to understanding human movement control as well as augmenting or assisting human movement.

### 4.2. Influence of Load Forces and Task Configurations on the Dynamical Synergies

Previous studies have found that the shared force patterns were affected by the object configurations such as texture, size and shape of the objects and the power of grasp involved [29,30]. Here, the dynamical synergies across four different weights were observed. Figure 6 illustrates the contribution of each hand area or finger in the first four synergies. The first two synergies shared common load force zones and the dominant areas were located at the thumb and middle finger. High correlations across four weights were found for the first two synergies. This may suggest that the first two dynamical synergies contain the functional basis for most hand dynamics across different weights. The dynamical synergies derived from spatial force maps are shown in Figure 6B. Similar to Figure 6A, the first two synergies shared similar force distribution among object weights, and the most common characteristics across varied weights are represented primarily on the middle finger, and secondarily on the thumb and lower palm. For higher-order synergies (third and above), the dominant load force zones differ for different weights. These could be subtle force adjustments that cannot be directly attributed to dominant force patterns observed across different object configurations or across different individuals. Nevertheless, the shared patterns of the first synergy suggest that the finger digits that contributed the most to the prehension forces are consistent across various weights and various prehension tasks, providing the probability of diverse types of grip force production using the same set of dynamical synergies. This may suggest that the most significant synergies accounted for variability across various prehensions; the higher-order synergies contain subtle information attributed to specific prehension tasks and are still very much helpful in fine control of hand prehensions [22]. Furthermore, the dynamical synergies derived from both averaged force vectors as well as from the spatial force maps, are consistent with these findings. The first and second dynamical synergies are similar among most subjects as shown in Figure 7 and Figure 8.

### 4.3. Biomechanical Constraints and Neural Control

During the steady-hold period, the gravitational force of the object can be considered as equal and opposite to the frictional force, which is directly related to the contact material and the normal forces. The contact material is consistent throughout all the objects and weights in this study. However, the DoFs of the contact forces are still too large. The DoFs of the contact forces are defined as the number of finger joints, or in this case, the contact areas involved in a certain grasp [24]. There are redundant DoFs participating in a grasping task that increase the computational load of control by the CNS [3]. Additionally, the DoFs involved in various types of hand prehension are not the same. Theoretically, each DoF is independently controlled, and it is possible to realize this when the grasp is simple, e.g., a pinch grasp with contact forces at the tips of thumb and index finger only. However, both direction and magnitude of the contact force cannot be determined and controlled when there are a large number of DoFs involved, especially when subjected to biomechanical constraints. When we move one finger of our hand to exert force, other fingers also generate force. This phenomenon has been defined and observed as enslaving or lack of individuation [31]. This biomechanical constraint could reduce redundant DoFs and thus reduce the load on the CNS, and studies have shown that there could be neural origins of enslaving [32]. Santello et al. proposed hierarchical control to explain the organization of contact forces and force synergies in grasping [16], and that the CNS might not control all parameters of grasp individually. Such hierarchical human-inspired control strategies were adapted in prosthetic and robotic hands [33,34]. Thus, the coordination patterns observed across the contact forces in this experiment might be imposed by the higher-level neural controls. In [35] it was observed that the force patterns across the fingers contributing to one grasping task can be extended to other grasping tasks. Thus, neural control combined with the biomechanical constraints may contribute to the coordination observed as the dynamical synergies in this experiment. However, it is difficult to estimate their individual contributions to the generation of these synergies [36]. It is possible that the first two synergies (Figure 6) that remained consistent across different grasps and different weights (and different subjects as shown in Figure 7 and Figure 8) mostly reflect the contribution of biomechanical constraints because they were consistent irrespective of task constraints. If this is true, then the higher-order synergies may mostly reflect neural control. How the biomechanical constraints and neural control affect synergies still needs further investigation.

### 4.4. Potential Applications in the Design and Control of Hand Exoskeletons

Hand synergies have been successfully used to enhance the control of robotic and prosthetic hands [7,11,12,13,14,37,38]. Thus, the low-dimensional DoFs of hand force control using dynamical synergies can provide an optimal approach to reduce the complexity of the design and control of hand exoskeletons. This will be evaluated on our previously designed hand exoskeleton with embedded synergies (HEXOES) [12] in the near future. The integration of dynamical synergies combined with postural [6] and kinematic synergies [12,26] can potentially improve the control. The act of hand prehension involves reaching, grasping, holding and manipulating. While the reaching and grasping phase can be represented by the combination of a few kinematic/postural synergies, the holding phase can be represented by a few dynamical synergies (Figure 3 and Figure 6). The manipulation phase will involve a combination of these synergies that will be explored in the near future. It has been shown the force-closure property of grasps strongly depends on the postural synergies selected to realize the control [39]. Therefore, dynamical synergies combined with postural and kinematic synergies can complete a grasp-to-hold movement and solve the above limitations and possibly provide accurate feedforward force control as well as sensory feedback to improve grasp performance [16]. As detailed in [33] the design and control of a prosthesis or an exoskeleton involves: (1) a mechatronic device with sensors and actuators (2) a decoder that interprets human intent and (3) a control law that translates all inputs to desired movements. Bioinspired controllers were only recently introduced in the design and control. Based on our experience [12,40,41,42] we believe that the dynamical synergies can be introduced in low-level control that helps in automatic adjustment of initial postures and grasp forces.

The dimensionality reduction through synergies is not only observed in the motor domain, but also appears to apply in the sensory domain by reducing sensory inputs to a low-dimensional space [16]. The sensory feedback to either the controllers or directly to the users is essential in force control during a grasp [43]. The sensorimotor synergies not only contribute to simplifying the control problem but also to reducing ambiguous interpretation of sensory information from touch [38]. According to the diverse synergy patterns discovered in this paper, the similarity of the first two dynamical synergies across different object weights (Figure 6) and the consistency among different individuals (Figure 7 and Figure 8) it is possible to reduce the sensory inputs depending on the primary load force areas. Furthermore, the derivations of higher-order synergies that seem to differ across subjects and weights can preserve anatomical and physiological differences in grasps across individuals, which can assist to accomplish subtleties in precise grip force control. 

### 4.5. Limitations and Future Directions

To realize the utilization of dynamical synergies into application, understanding the influence of the task environment (shape, size, texture, and weight of the objects) is essential. The objects selected in this study represent four commonly used grasp types from ADLs and they were tested with four different weights. Including more objects and grasp types can help to verify the consistency of the synergies and generalizability of the results. Additionally, hand size can affect the contact forces recorded by the sensors. We will account for hand sizes of the subjects in our follow-up studies. We also will test our methods in the datasets that were already collected, e.g., [20]. Furthermore, as natural hand is equipped with about 17,000 mechanoreceptors [33] that offer tremendous spatiotemporal tactile sensation, we will use high resolution gloves (548 sensors in this new tactile glove [19] vs. 349 sensors in Tekscan GripGlove used in this study) to determine what is the necessary resolution to perform a grasp. Object manipulation is another aspect that will be considered in our future studies, as it affects the grasp posture [33].

## 5. Conclusions

This study demonstrated the reconstruction of the hand prehension forces using only a few dynamical synergies. The consistency of lower-order synergies provides shared functionality of contact force strategy across different object weights as well as across different individuals, while the specificity of higher-order synergies could provide the user with an increased sense of ownership and may also assist in finer sensorimotor control. The dynamical synergies are expected to help reduce both grip force and sensory input variables involved in complex sensorimotor control conducted effortlessly by the CNS. We aim to test this in the near future in a hand exoskeleton with embedded synergies (HEXOES) developed in our lab [12]. Questions remain unanswered as to how the biomechanical constraints and the neural control contribute to the development of synergies and how kinematic and dynamic synergies can be integrated efficiently to improve the performance of prosthetics and exoskeletons. Overall, these dynamical synergies could enhance our understanding of how the CNS might implement a synergistic control of hand prehension. 

## Figures and Tables

**Figure 1 sensors-22-04177-f001:**
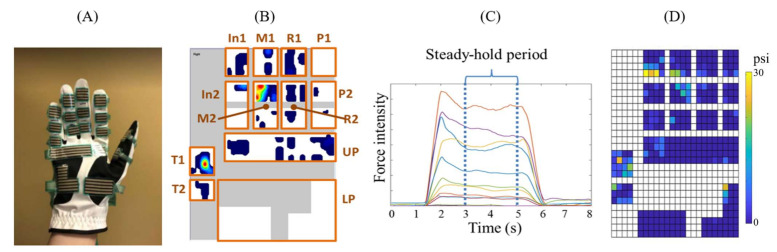
(**A**) GripGlove (Tekscan, Boston, MA, USA) with 12 sensors to capture the contact forces in the 12 hand areas explained as follows. (**B**) 12 areas from where hand forces were recorded, including upper palm (UP), lower palm (LP), distal and proximal phalanx of thumb (T1 and T2), distal phalanges of index, middle, ring and pinky fingers (In1, M1, R1, P1) and middle-proximal phalanges of four fingers (In2, M2, R2, P2). (**C**) Recorded average forces during the steady-hold period. Colors indicate the 12 areas. (**D**) Recorded spatial force maps pixelated corresponding to the 12 hand areas.

**Figure 2 sensors-22-04177-f002:**
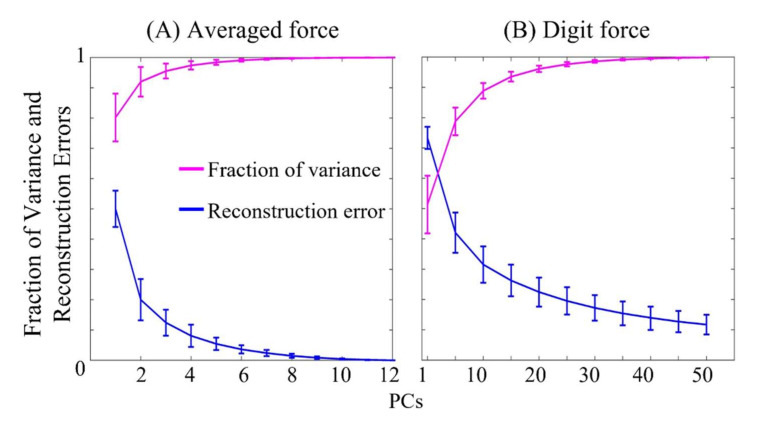
The fraction of the variance accounted for and the reconstruction errors were calculated using the top-order synergies. The mean and standard deviation was calculated across ten subjects. (**A**) 12 synergies were extracted from the averaged force vectors and (**B**) 361 synergies extracted from spatial force maps (only the first 50 synergies) are shown here.

**Figure 3 sensors-22-04177-f003:**
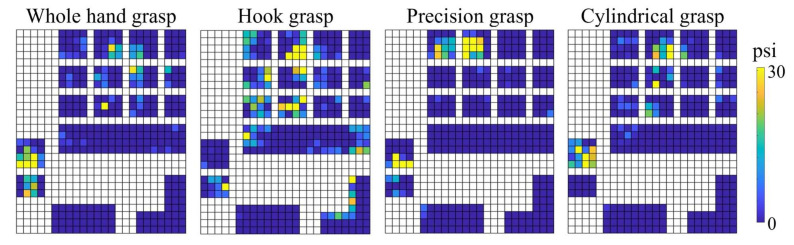
Force distribution of four grasp types of Subject 3 (for object weight 170 g). The whole hand grasp (ball) and cylindrical grasp (water bottle) are mostly concentrated at the fingers. The dominant fingers for precision grasp (bottle cap) are the tips of the thumb, index and middle fingers. Hook grasp (door handle) requires both palm, thumb and fingers. The scale psi indicates pound force per square inch.

**Figure 4 sensors-22-04177-f004:**
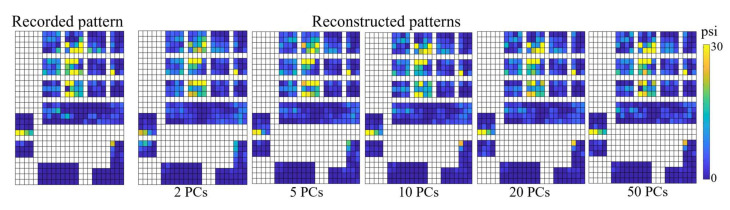
Comparison between the recorded and reconstructed spatial force maps using different numbers of PCs. The patterns are taken from the whole hand grasp of Subject 6 for object weight 170 g. The reconstructed patterns become more explicit and accurate when more synergies are recruited. Using only a few synergies were not able to accurately reconstruct the force patterns but were able to capture the dominant grasp force characteristics. The scale psi indicates pound force per square inch.

**Figure 5 sensors-22-04177-f005:**
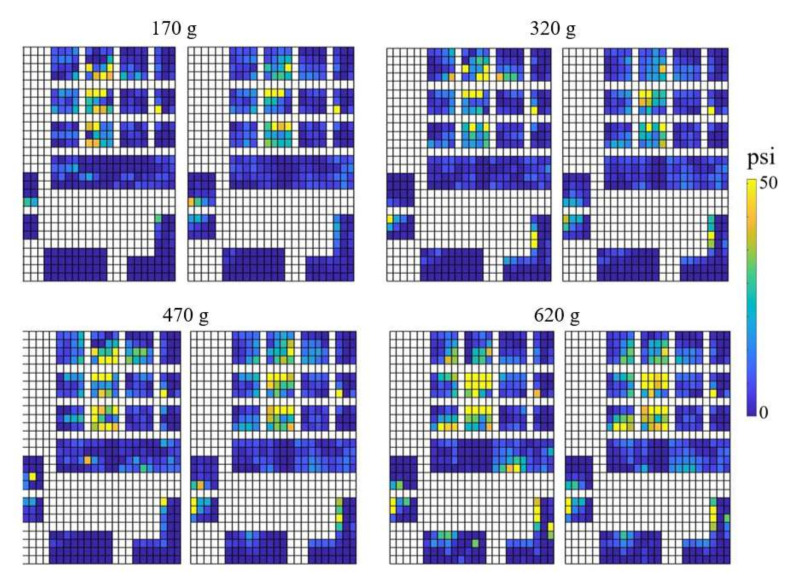
Reconstructed spatial force maps for whole hand grasp for four different weights using the first two synergies for Subject 6. Most of the force intensity is concentrated at the thumb and middle fingertips when holding the object at 170 g. The reconstructed maps using only two synergies could successfully capture the dominant force zones. When extra weights are added to the objects, the force zone enlarges, and fingers involved in the grasp increase. The color bar indicates the scale of the contact forces. Increase in the color bar scale indicates increase in the forces due to increase in the weights. The scale psi indicates pound force per square inch.

**Figure 6 sensors-22-04177-f006:**
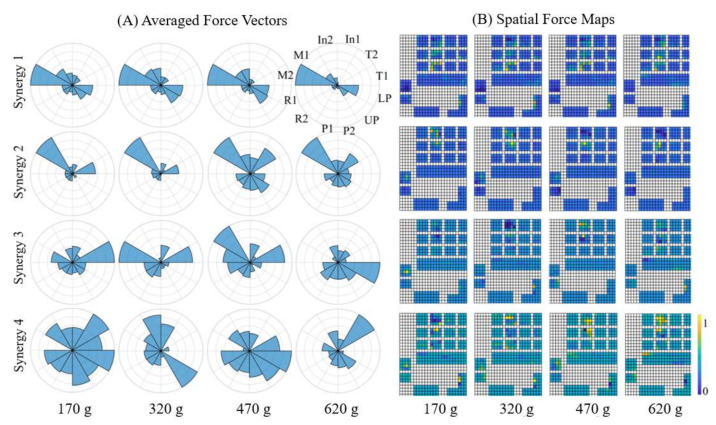
(**A**) Dynamical synergies extracted from averaged force vectors of Subject 8. The first and second synergies share the common load force zones regardless of the object weight. The middle digits (M1 and M2) carried the most dominant force characteristics; thumb and palm also contributed to sharing the force strategies. High correlations were observed among four weights. (**B**) Dynamical synergies extracted from spatial force maps of Subject 8. The first two synergies share the common load force zones and there is variability across synergies. The color bar indicates the normalized scale of forces obtained from principal components.

**Figure 7 sensors-22-04177-f007:**
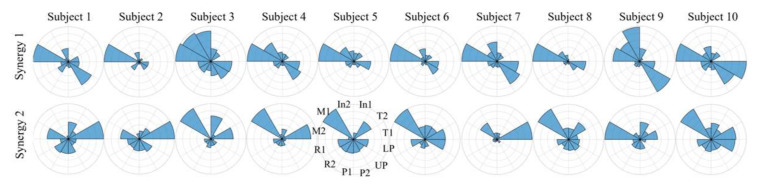
The first two dynamical synergies derived from averaged force vectors (for object weight 620 g) from all ten subjects. The first two dynamical synergies share the most common load force areas and are consistent across the subjects. The middle finger (in Synergy 1) and thumb (in Synergy 2) are the most used as captured by these two synergies. The other common load force areas are the index finger and palm.

**Figure 8 sensors-22-04177-f008:**
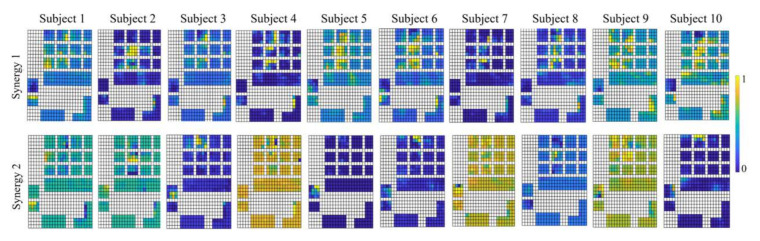
The first two dynamical synergies derived from digital force patterns (object weight 620 g) from all ten subjects. The dominant load force areas are concentrated on the middle finger and thumb. The first synergy carries a consistent force distribution among all the subjects, while the second synergy contains both common force information as well as individual variance. The color bar indicates the normalized scale of forces obtained from principal components.

**Table 1 sensors-22-04177-t001:** Reconstruction error (mean(std)) of four weights from 10 subjects with.

	1	2	3	4	5	6	7	8	9	10
170 g	0.13 (0.12)	0.05 (0.07)	0.04 (0.04)	0.07 (0.08)	0.09 (0.12)	0.06 (0.05)	0.12 (0.12)	0.03 (0.03)	0.09 (0.06)	0.03 (0.04)
320 g	0.1 (0.06)	0.06 (0.07)	0.03 (0.04)	0.04 (0.07)	0.12 (0.17)	0.05 (0.06)	0.07 (0.09)	0.04 (0.04)	0.1 (0.08)	0.04 (0.06)
470 g	0.09 (0.08)	0.07 (0.07)	0.04 (0.05)	0.05 (0.05)	0.12 (0.15)	0.04 (0.05)	0.06 (0.05)	0.05 (0.04)	0.16 (0.15)	0.06 (0.1)
620 g	0.17 (0.14)	7.5 (9.1)	0.08 (0.06)	0.04 (0.05)	0.15 (0.16)	0.07 (0.08)	0.08 (0.07)	0.06 (0.08)	0.14 (0.17)	0.07 (0.09)

## Data Availability

Not applicable.

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
