# Peer review of "Dynamical Synergies of Multidigit Hand Prehension"

_sensors, 2022, doi:10.3390/s22114177_

Round 1

Reviewer 1 Report

The paper describes an interesting research about contact force distribution in the human hand and extracts dynamical synergies when a set of objects with different weights are grasped using different grasp types. Despite the set of objects being limited, the study is well conducted and presented and constitutes an interesting contribution. However, there are several ideas that can improve the current version of the paper:

  • I put the attention of the authors on the paper by Cepria-Bernal&Pérez-González, 2021 (https://pubmed.ncbi.nlm.nih.gov/33917212/) and also in that of Sundaram et al., 2019 (https://www.nature.com/articles/s41586-019-1234-z). Given the similar approach used in those previous studies, maybe you should mention them in the introduction, as previous related research, and also in the discussion, where you mention the absence of previous research on spatial force maps (line 215), which is not correct.
  • The hand size could affect the results, because of differences in glove fitting. You should supply information about the hand size of the subjects.
  • Units for the scale in figure 3 and similar figures, must be specified.
  • In the discussion and references to figures 6 and 7, you explain that the middle finger and thumb are the most recruited areas, but in the figures of Synergy 1 it seems that the palm (LP and UP) is loaded more. Please revise that labels for the zones are OK in these figures and revise explanations accordingly.
  • You should indicate the limitations of the study in the discussion. One is the limited number of objects to reach definitive conclusiones. You should also comment about the problems observed during measurements with the sensor or glove fitting.

Minor comments:

  • Correct typo “?contains” in line 126.
  • Revise “is can be” in line 324
  • Revise “as shown in spatial force maps as shown” in line 330
  • Revise wording in the last sentence of the conclusion.

Author Response

Please see below our responses.

Reviewer 2 Report

In this work, the authors investigate the contribution of dynamical synergies, and coordinated spatial patterns of contact forces to the contact forces in a grasp trying to understand if the dynamical synergies could potentially capture feedforward and feedback mechanisms. 

Major
Lines 61-64: The authors wrote "several studies" but cited just one of them. I suggest they amplify the state of the art on this part.
Lines 64-70: The authors assumed the non-solved problem regarding "the cues from the environment (shape, size, texture, and weight) for optimal control of multiple DoF at hand (muscles and joints)". I suggest the authors read the book "The Human Hand as an Inspiration for Robot Hand Development" written by Ravi Balasubramanian and Veronica J. Santos to deeply understand human hand behaviour.

Rationale and aim are not clear, then I suggest the authors to better explain them.

Paragraphs 4.3 and 4.4: For paragraphs, I recommend reading the article:
Hierarchical Human-Inspired Control Strategies for Prosthetic Hands
In which many aspects not clear to the authors are explained but can, through the bibliography, deepen the neurophysiological aspects of prehension. Moreover, Paragraph 4.4 looks like a Future work since the authors did not test anything on prostheses.

Both in Introduction and in Conclusions, it is not clear what the authors discovered and how it is useful in prosthetics or robotics field. In particular, it is evident the authors have a little experience in the control of prosthetic hands and they do not know what is important in a control law for an amputee. I suggest the authors deeply study the field of the control strategy for prosthetic hands to find a link with their studies. 

This study does not clarify the improvement of the state of the art because it lacks the information. It is mandatory to add other information (other articles) and highlight the pros and cons and formulate the question related to the study.

Conclusions are not supported by all the data but it is focused on robotics and prosthetics, fields not deeply by the authors.

Line 27: ADLs
Line 35: DoFs
Line 83: to specify the kind of the objects
Line 84: g instead of grams
Line 126: control "?contains"
Paragraph 2.3 does not have any citations, I suggest the authors to revision it.
Line 308: Paragraph 4.4 

Author Response

Please see below for our responses.

Round 2

Reviewer 2 Report

The authors answered satisfactorily my questions. In this form the article is acceptable.

Author Response

Thank you very much for your comments that has helped in improving the manuscript immensely. From further comments from Academic Editor we have improved the writing, grammar, spellings through out the manuscript.Thank you again for your comments and you have been added in acknowledgements.